# Copper Modulates Adult Neurogenesis in Brain Subventricular Zone

**DOI:** 10.3390/ijms23179888

**Published:** 2022-08-31

**Authors:** Luke L. Liu, Richard M. van Rijn, Wei Zheng

**Affiliations:** 1School of Health Sciences, Purdue University, West Lafayette, IN 47907, USA; 2Department of Medicinal Chemistry and Molecular Pharmacology, College of Pharmacy, West Lafayette, IN 47907, USA

**Keywords:** copper, subventricular zone, adult neurogenesis, neurosphere, D-penicillamine, intracerebroventricular infusion

## Abstract

The subventricular zone (SVZ) in lateral ventricles is the largest neurogenic region in adult brain containing high amounts of copper (Cu). This study aims to define the role of Cu in adult neurogenesis by chelating labile Cu ions using a well-established Cu chelator D-Penicillamine (D-Pen). A neurosphere model derived from adult mouse SVZ tissues was established and characterized for its functionality with regards to neural stem/progenitor cells (NSPCs). Applying D-Pen in cultured neurospheres significantly reduced intracellular Cu levels and reversed the Cu-induced suppression of NSPC’s differentiation and migration. An in vivo intracerebroventricular (ICV) infusion model was subsequently established to infuse D-Pen directly into the lateral ventricle. Metal analyses revealed a selective reduction of Cu in SVZ by 13.1% (*p* = 0.19) and 21.4% (*p* < 0.05) following D-Pen infusions at low (0.075 μg/h) and high (0.75 μg/h) doses for 28 days, respectively, compared to saline-infused controls. Immunohistochemical studies revealed that the 7-day, low-dose D-Pen infusion significantly increased Ki67(+)/Nestin(+) cell counts in SVZ by 28% (*p* < 0.05). Quantification of BrdU(+)/doublecortin (DCX)(+) newborn neuroblasts in the rostral migration stream (RMS) and olfactory bulb (OB) further revealed that the short-term, low-dose D-Pen infusion, as compared with saline-infused controls, resulted in more newborn neuroblasts in OB, while the high-dose D-Pen infusion showed fewer newborn neuroblasts in OB but with more arrested in the RMS. Long-term (28-day) infusion revealed similar outcomes. The qPCR data from neurosphere experiments revealed altered expressions of mRNAs encoding key proteins known to regulate SVZ adult neurogenesis, including, but not limited to, Shh, Dlx2, and Slit1, in response to the changed Cu level in neurospheres. Further immunohistochemical data indicated that Cu chelation also altered the expression of high-affinity copper uptake protein 1 (CTR1) and metallothionein-3 (MT3) in the SVZ as well as CTR1 in the choroid plexus, a tissue regulating brain Cu homeostasis. Taken together, this study provides first-hand evidence that a high Cu level in SVZ appears likely to maintain the stability of adult neurogenesis in this neurogenic zone.

## 1. Introduction

The subventricular zone (SVZ) is the largest neurogenic region in adult mammalian brain. It lines the wall of lateral ventricles, and harbors adult neural stem/progenitor cells (NSPCs). The NSPCs in the SVZ remain largely quiescent; but upon activation, NSPCs differentiate into neuroblasts and enter the rostral migratory stream (RMS), migrating toward their destination brain regions. The olfactory bulb (OB) is the primary destination for these SVZ-derived migratory neuroblasts [1]. Within the OB, newly arrived neuroblasts further differentiate into granule cells and periglomerular cells in the granule cell layer (GCL) and glomerular layer (GL), respectively. Importantly, these two types of replenishable interneurons by SVZ neurogenesis regulate the normal olfactory function as they form synapses with the OB projection neurons that receive signals from the olfactory sensory neurons in the olfactory epithelium. Through these interneuron connections, the signal is further projected to the cortex for olfactory sensation. Thus, any adverse alterations in the SVZ adult neurogenesis can lead to changes in new neuron supply in the OB, ultimately impairing the smell function [1]. 

Anosmia (i.e., partial or full loss of smell) is a frequently reported non-specific symptom that precedes by years the hallmark neuropathology in Alzheimer’s diseases (AD) and Parkinson’s disease (PD) [2]. Evidence in the literature has identified a compromised SVZ adult neurogenesis in patients and animal models of AD and PD [3,4,5,6,7,8]. However, the mechanisms by which the SVZ adult neurogenesis contributes to the neurodegeneration is largely unknown. Interestingly, metal dyshomeostasis, especially these essential elements such as copper (Cu), zinc (Zn), iron (Fe), and manganese (Mn), has been associated with the aberrant SVZ and ensuing neurodegenerative alterations. For example, reports suggest that dyshomeostasis of Cu, Zn, Fe, and Mn in the brain not only accelerates the neurodegeneration [9,10,11,12,13,14], but also causes disrupted SVZ adult neurogenesis [15,16,17,18]. Recent studies even proposed that in COVID-19 patients observed with abnormally high metal concentrations, an aberrantly activated SVZ adult neurogenesis may cause a grey matter volume enlargement in OB [19,20]. However, the relationship between patients’ metal dyshomeostasis and abnormal SVZ adult neurogenesis has yet to be elucidated.

Among the essential trace elements, several lines of evidence support Cu as a suppressor of SVZ adult neurogenesis. First, reports by both synchrotron X-ray fluorescence (XRF) and atomic absorption spectrometry (AAS) discovered a selective Cu enrichment in the SVZ in an age-dependent manner [21,22]. Cu was concentrated within GFAP(+) cells in the SVZ, which represent largely NSPCs [21,23]. Second, toxicological studies showed that a decreased Cu level in the SVZ after Mn overexposure abnormally increased NSPC proliferation, albeit with a declined new neuron production at OB; the latter seemed likely owing to an arrested migration of newborn neuroblasts in RMS [16,17]. Third, directly delivering Cu ions to the cerebrospinal fluid (CSF) by intracerebroventricular (ICV) injections apparently suppressed the NSPC proliferation in the SVZ [16]. Finally, a recent report by this lab modeling systemic Cu overload in rats further revealed that the SVZ adult neurogenesis was significantly downregulated in brains of Cu-overladed animals [24]. These observations prompted us to propose a Cu-dependent regulatory mechanism in the SVZ, where a naturally high Cu level in the SVZ may suppress the adult neurogenesis in a state of “switching-off”, whereas a reduced SVZ Cu may “switch on” the neurogenesis, thus increasing the proliferation, migration, and differentiation of NSPCs along the SVZ-RMS-OB axis. This hypothesis became the subject of this study for experimental testing.

To investigate Cu participation in biological processes, there is a need to use the agents that specifically modulate Cu homeostasis. D-penicillamine (D-Pen), a selective Cu chelating agent, has been extensively used in research as well as in clinics, and proven to be capable of effectively removing Cu ions in body fluids and lowering Cu burden in patients [25,26]. D-Pen (trade name Cuprimine) has been clinically used to treat Wilson’s disease, an inherited genetic disorder that causes abnormal Cu accumulation in the body and leads to severe brain and hepatic symptoms [26]. D-Pen can form a stable, water-soluble complex with Cu intracellularly and extracellularly, allowing a rapid urinary excretion of Cu from the body [26]. Thus, D-Pen is one of the most frequently used Cu-chelating agents in Cu biological and toxicological research.

The main purpose of this study was to understand the role of Cu in regulating adult neurogenesis in the SVZ. To reach this goal, we developed an SVZ-derived in vitro neurosphere model and used the unique Cu chelating agent D-Pen to elucidate how Cu status determined the proliferation, migration, and differentiation of NPSCs in neurospheres. We then extended the research in live animals by establishing an in situ intracerebroventricular (ICV) D-Pen infusion model, followed by tracing the newly generated neuroblasts/neurons in the SVZ-RMS-OB axis using immunohistochemistry (IHC) approach. Finally, we investigated the expression levels of Cu regulatory proteins in the SVZ and choroid plexus tissues using IHC. The results of this study provide the firsthand evidence to support a regulatory role of Cu in adult neurogenesis in the SVZ. 

## 2. Results

### 2.1. Establishment and Characterization of Neurospheres Derived from Adult SVZ In Vitro

In neurospheres formed with relatively identical sizes (Figure 1D), the immunofluorescent staining showed a robust number of BrdU(+) proliferating cells (Figure 1E–G), with the BrdU signals overlapping with Nestin (Figure 1E), indicating an abundance of NSPCs in neurospheres. However, the fluorescent signals for doublecortin (DCX) and NeuN, two markers of neuronal differentiation, were low to none (Figure 1F,G), suggesting that the neurospheres generated by our current method were primarily enriched with NSPCs capable of further proliferation and differentiation, but not the differentiated cell types. Analyzing the cell migration from the core indicated that after 7 days in culture, cells not only migrated outward covering a larger area, but also differentiated into NeuN(+) mature neurons (Figure 1H). Since adult NSPCs cultured in vitro are reportedly susceptible of apoptosis [27], we examined the apoptotic level in our neurospheres by using an apoptosis detection kit. The data in Figure 1I showed negligible apoptotic signals, while the positive control neurospheres treated with 1.6 mM H_2_O_2_ for 30 min displayed strong apoptotic signals (Figure 1J). These data suggest that the neurospheres established by our experimental workflow possessed the properties favorable for further investigation detailed below.

### 2.2. Cu Dose-Dependent Suppression of Neurosphere Growth In Vitro

Using the in vitro SVZ-derived neurosphere model, we investigated the impact of Cu treatment on neurosphere migration as well as their differentiation into NeuN(+) neurons. Data presented in Figure 2A showed a Cu dose-dependent suppression of NPSC differentiation; the presence of Cu in the culture medium at 10.0 and 100.0 µM significantly reduced the NeuN(+) cells by 22% (*p* < 0.05) and 38% (*p* < 0.001), respectively, compared to controls (Figure 2B). The similar Cu impact was also observed in the migration area, where Cu at 10.0 and 100.0 µM reduced the migration area by 21% (*p* < 0.05) and 33% (*p* < 0.001), respectively (Figure 2C). In addition, we observed that the NeuN signal, which is supposedly localized in nucleus [28], translocated to the peri-nuclear area, especially when the neurospheres were treated with high Cu concentrations (Figure 2A). Thus, these observations established that Cu dose-dependently suppressed the SVZ adult neurogenesis in vitro.

### 2.3. Chelation of Cu by D-Pen Revised Cu Suppression of Neurosphere Growth In Vitro

#### 2.3.1. Treatment with D-Pen Alone

To prove the role of Cu in regulating adult neurogenesis, we used a Cu chelating agent D-Pen to examine the growth of neurospheres following Cu chelation treatment. As the first step toward D-Pen chelation, we investigated whether treatment by D-Pen itself would affect the neurosphere growth and differentiation. Data in Figure 3 demonstrated that the presence of D-Pen at 20 µM in culture medium significantly increased the NeuN(+) cell counts by 14% (*p* < 0.05) (Figure 3B); it also increased the cell migration area by 18% (*p* < 0.05) (Figure 3A,C). Noticeably, D-Pen at 50 µM caused the largest migration by 21% (*p* < 0.05) as well as the mild NeuN signal translocation from nuclei to cytosol (Figure 3A). At 100.0 µM D-Pen, a widespread NeuN translocation was observed. Thus, we decided to use 20-µM D-Pen for subsequent chelation experiments. 

#### 2.3.2. Treatment with Both Cu and D-Pen

To verify the effectiveness of D-Pen in reducing intracellular Cu levels, we used Phen Green, a fluorescent Cu sensor, to estimate the intracellular Cu burden. Fluorescent imaging by Phen Green showed a strong green fluorescence in the control neurospheres, while incubation with 10-µM Cu significantly quenched the fluorescent signals (Figure 4A). The presence of D-Pen in culture medium greatly restored the quenched fluorescent signals, indicating a reduced cellular Cu level after D-Pen chelation (Figure 4A). Further estimation of intracellular Cu burden by Phen Green imaging revealed that D-Pen supplement at 20 µM efficiently lowered the Cu overload induced by Cu (10 µM) exposure: (F_ct_ − F)/F_ct_ values were almost brought to the baseline level by D-Pen supplementation (Figure 4B). 

The neurospheres were then treated with 10 µM Cu alone or Cu with 20 µM D-Pen to investigate whether the presence of D-Pen alleviated the Cu-suppressed neurosphere growth. Data in Figure 4C showed that D-Pen intervention reversed the Cu-induced neurogenesis suppression as evidenced by the revised NeuN counts per neurosphere (Figure 4D) and the nearly normal migration areas (Figure 4E). Noticeably also, NeuN translocation to the peri-nuclear regions by Cu overload was largely alleviated by D-Pen. These in vitro data implied that Cu was required by adult neurogenesis; excessive Cu, however, was capable of suppressing adult neurogenesis in SVZ.

### 2.4. Expressions of mRNAs Encoding Critical Regulatory Proteins in Neurospheres as Affected by Cu Status

To understand the mechanisms whereby Cu status modulates neurosphere growth, we used qPCR to explore the changes of a battery of regulatory proteins in the process. The first group was the proteins involved in Cu transport. Among the tested Cu transporters (Ctr1, Dmt1, Atp7a and Atp7b), only Atp7b was significantly upregulated by 85% (*p* < 0.05) following 10 µM Cu treatment; this upregulation, however, was effectively brought to normal by 20 µM D-Pen chelation treatment (Figure 5A). The second group pertained to the intracellular Cu binding proteins, metallothioneins (MTs). Of three MTs, incubation with 10 µM Cu upregulated expression of Mt2 by 35% (*p* < 0.05); surprisingly, D-Pen supplementation increased its expression even more by 49% (*p* < 0.01) (Figure 5B). 

A much greater change became evident in mRNAs pertinent to SVZ adult neurogenesis in the third testing group, including β-catenin, Notch1, Shh, and Dlx2, all of which play critical regulatory roles in adult NSPCs in the SVZ [29]. Treatment with 10-µM Cu in culture medium significantly altered their expression (Figure 5C). Although D-Pen chelation brought the expression levels of Shh and Dlx2 back to normal, it failed to reverse the decreased expression of β-catenin and Notch1 after Cu treatment. The fourth group included the gene products in modulating neuroblast migration in the RMS [30]. Incubation with Cu significantly reduced the mRNA expression of Slit1, Slit2 and Robo1, which was partly alleviated by D-Pen chelation (Figure 5D). 

These data suggest that adult neurogenesis-related markers in SVZ appeared to be more responsive to Cu treatment than those of Cu transporters or binding proteins. Chelating Cu by D-Pen allowed the partial recovery of the mRNAs encoding neurogenesis modulators.

### 2.5. Effect of ICV Infusion of D-Pen on Cu Levels in Selected Brain Regions

To verify the observed Cu participation in adult neurogenesis from our in vitro neurosphere D-Pen experiments, we established an in situ ICV infusion model in mice for short-term (7 days) and long-term (28 days) D-Pen treatments (Figure 6A). Following long-term D-Pen ICV infusion, the AAS was used to quantify Cu concentrations in multiple brain regions, i.e., SVZ, RMS, OB, hippocampus, frontal cortex, and striatum. In the SVZ, the high-dose D-Pen infusion significantly decreased Cu levels by 21.4% (30.6 ± 6.2 µg/g; *p* < 0.05), compared to the saline infused controls (38.9 ± 2.7 µg/g). D-Pen infusion at the high dose also decreased the Cu levels by 24.2% (*p* = 0.10) and 38.0% (*p* = 0.11) in hippocampus and frontal cortex, respectively, compared to the saline controls, although the levels of Cu decreases did not reach statistical significance. Cu levels in the RMS, OB, hippocampus, and frontal cortex, however, did not change significantly after D-Pen infusion (Figure 6B). Understandably, the SVZ, being a part of the ventricular wall directly encountering the ventricular CSF, would sense the impact of D-Pen (which was delivered directly to the brain ventricle) more readily than other brain regions. Interestingly, D-Pen infusion elevated the Cu concentration in the striatum by 45.1% (*p* = 0.19) and 82.1% (*p* < 0.05) after the low-dose and high-dose D-Pen treatment, respectively. These data suggest that ICV infusion of D-Pen was able to decrease the Cu level in the SVZ while spared other brain regions.

### 2.6. Impact of Short-Term ICV Infusion of D-Pen on the SVZ-RMS-OB Axis

To understand the role of Cu in regulating adult neurogenesis alongside of the SVZ-RMS-OB axis, we infused D-Pen into brain ventricle and quantify NPSC’s proliferation, migration, and differentiation using specific markers for each process. Short-term (7-day) D-Pen infusion significantly increased the number of Ki67(+)-Nestin(+) cells (i.e., actively proliferating NSPCs) (1664 ± 172 per brain) in the SVZ in D-Pen (low-dose) group, as compared to saline controls (1220 ± 118 per brain) (*p* < 0.05), but the high-dose D-Pen treatment did not cause significant changes (1456 ± 102 per brain) (Figure 7A,B). Therefore, short-term D-Pen infusion at the low dose apparently stimulated the SVZ adult neurogenesis by upregulating the proliferation. 

Increased neuroblasts detected in the RMS likely signals an upregulated mobilization of NPSCs from SVZ. Interestingly, short-term D-Pen infusion at the high dose, but not low dose, led to an increased DCX(+)/BrdU(+) newborn neuroblasts in the RMS by sagittal section staining (Figure 7C,D). Counting these DCX(+)/BrdU(+) neuroblasts in the RMS revealed that short-term infusion of D-Pen at the high dose yielded significantly more migrating neuroblasts in RMS (477 ± 30 per brain, *p* < 0.001) than those in controls (257 ± 44 per brain, *p* < 0.001)) and in D-Pen (low-dose) group (244 ± 33 per brain, *p* < 0.001) (Figure 7D). Thus, it is likely that the high-dose D-Pen, but not low-dose D-Pen infusion, propelled the SVZ adult neurogenesis by generating more differentiated newborn neuroblasts and increasing their migration via RMS to the OB.

The OB is the destination of the newly generated neuroblasts in the SVZ. The imaging data in Figure 7E suggested the presence of DCX(+)/BrdU(+) newborn neuroblasts in the OB following short-term infusion. Further quantifying these cells revealed that the arrival of newborn neuroblasts in the OB was significantly higher in the low-dose D-Pen group (5580 ± 276 per brain) than that in saline controls (4564 ± 181 per brain) (*p* < 0.05). However, the high-dose D-Pen infusion caused a significantly lower neuroblast arrival in the OB (3716 ± 352 per brain, *p* < 0.05) as compared to controls (Figure 7F). Taken together, the data in Figure 7 appeared to suggest that the low-dose D-Pen infusion stimulated the proliferation of newborn NPSCs in the SVZ (Figure 7B) and propelled the neuroblast migration towards OB (Figure 7F), whereas the high-dose D-Pen infusion may arrest these neuroblasts in the RMS. Since one cycle of SVZ adult neurogenesis takes approximately one month, a thorough examination of the SVZ-RMS-OB activity beyond the 7-day paradigm became necessary.

### 2.7. Impact of Long-Term ICV Infusion of D-Pen on the SVZ-RMS-OB Axis

#### 2.7.1. Changes in SVZ NSPC Pool

Adult NSPCs residing in the SVZ are the source for descendent interneurons in the OB. Given that SVZ harbors multiple types/stages of cells, we used CD133(+)/GFAP(+) cells to image and quantify neural stem cells (NSCs), and Nestin to differentiate the activated NSCs (aNSCs) from the quiescent NSCs (qNSCs) and to calculate the ratios of aNSCs to qNSCs along the lateral ventricular wall [31,32]. Representative images in Figure 8A showed robust signals of GFAP, CD133, and Nestin. An asterisk-marked aNSC and a pound-marked qNSC in a saline-infused SVZ (Figure 8A, upper right panel) were further magnified in Figure 8B and C, respectively, to reveal the cell identities. Of note, unlike the GFAP and Nestin localized in the cytosol, CD133 was found “floated” in the CSF, as previously reported [31]. 

Cell counting results, while revealing no changes in NSCs by D-Pen (low) (25.0 ± 4.4 per mm LV wall), showed a 46.2% decline in NSCs by D-Pen (high) infusion (14.0 ± 2.6 per mm LV wall, *p* < 0.05), in comparison to the saline infusion group (26.0 ± 3.6 per mm LV wall) (Figure 8D). Furthermore, quantification of aNSCs demonstrated that D-Pen (high) infusion greatly inhibited the aNSC number (2.5 ± 0.5 per mm LV wall) by 58.3% compared to saline (6.0 ± 1.3 per mm LV wall) (*p* < 0.05), although the increase by D-Pen (low) (8.3 ± 1.2 per mm LV wall) was not statistically significant (*p* = 0.08) (Figure 8E). Interestingly, the aNSC/qNSC ratio revealed that D-Pen (low) infusion upregulated the ratio of aNSCs to qNSCs in SVZ (50.3% ± 3.2%), as compared to the saline (30.0% ± 5.3%); but the ratio remained unchanged following D-Pen (high) treatment (23.7% ± 8.4%) (Figure 8F). Overall, these data suggest that the long-term, low-dose D-Pen infusion likely stimulated the SVZ adult neurogenesis without depleting the NSPC pool, while the high-dose D-Pen infusion may partially compromise the long-term repopulating potential of this NSPC pool.

#### 2.7.2. Changes in Newly Generated Mature Neurons in the OB

The newly generated NeuN(+)/BrdU(+) mature neurons arrive in the OB in two distinct areas, i.e., granule cell layer (GCL) and glomerular layer (GL) (Figure 9A). Long-term ICV infusion of D-Pen in mice revealed that, in the GCL, D-Pen (low) treatment significantly increased the NeuN(+)-BrdU(+) cells (4298 ± 449 per animal) by 37.8% compared to the saline group (3120 ± 394 per animal) (*p* < 0.05), while no changes were detected in the D-Pen (high) group (2966 ± 235 per animal). In the GL, however, no difference was observed among the three treatment groups. By combining the data from GCL and GL together, it was evident that D-Pen (low) treatment significantly increased the neurogenesis in OB (4868 ± 443 per animal) by 34.7% as compared with saline controls (3614 ± 380 per animal) (*p* < 0.05); but D-Pen (high) treatment did not show such a stimulatory effect. The findings suggest that D-Pen (low) treatment appeared likely to activate SVZ adult neurogenesis through the SVZ-RMS-OB axis.

#### 2.7.3. Expression of CTR1 in SVZ and Striatum

The SVZ possesses the highest Cu level in brain [21,22,33]. We set out to investigate if chelation of Cu by D-Pen may interfere the expression of proteins responsive to cellular Cu regulation in the SVZ. Data in Figure 10A showed that CTR1 was expressed in clusters in the SVZ, and its expression was much higher in the SVZ than in adjacent brain regions such as striatum. Magnified images highlighted the colocalization of CTR1 with Nestin and GFAP, suggesting an enriched expression in aNSCs (Figure 10A). However, since the three fluorescent signals for CTR1, GFAP, and Nestin were only partially overlapped, it was not entirely certain if the strong CTR1 expression was uniquely located in aNSCs. In addition, quantification of the average fluorescence intensity of SVZ CTR1 clusters did not reveal any differences among treatments (Figure 10B); but the striatal CTR1 was upregulated by 26.2% and 37.2% in the D-Pen (low) and D-Pen (high) group as compared to the controls, respectively (*p* < 0.05) (Figure 10D). Interestingly, we found that the average distance of CTR1 clusters to the CSF decreased by 39.6% and 62.8% by D-Pen (low) treatment (6.9 ± 0.77 µm) and D-Pen (high) treatment (4.3 ± 0.47 µm), respectively (*p* < 0.05), in comparison to control group (11.5 ± 1.9 µm) (Figure 10C). These data indicate that the D-Pen infusion did affect the expression of Cu transporter CTR1 in the SVZ and striatum.

#### 2.7.4. Expression of MT3 in SVZ and Striatum

MT3 is a major intracellular Cu-storage protein. Data from the current study found an enriched MT3 in the SVZ, but much lower expression in adjacent brain regions (Figure 11A). Similar to CTR1, MT3 signals only partially overlapped with those of GFAP and Nestin. Further quantification of MT3 fluorescent intensities in SVZ and striatum revealed that MT3 expression in SVZ was dose-dependently increased by 24.0% by low-dose D-Pen (*p* < 0.05) and 42.6% by high-dose D-Pen (high) (*p* < 0.01) (Figure 11B). However, striatal MT3 expression was not changed (Figure 11C). Thus, the long-term chelation of Cu in brain ventricles by D-Pen increased the expression of MT3 in the SVZ.

### 2.8. Impact of Long-Term ICV Infusion of D-Pen on CTR1 and MT3 Expression in the Choroid Plexus

The choroid plexus resides in brain ventricles immediately adjacent to the SVZ and regulates Cu homeostasis in the CSF. Long-term D-Pen infusion in mice significantly increased CTR1 expression in choroid plexus as compared to controls by 83.6% and 104.1% following D-Pen (low) and D-Pen (high) treatments, respectively (*p* < 0.001) (Figure 12A,C). In addition, D-Pen infusion altered the subcellular distribution of CTR1 for it being more concentrated toward the apical aspect of choroidal epithelial cells (CPECs), especially in the D-Pen (low) group (Figure 12A). As to MT3, D-Pen infusion neither elicited changes in its expression level nor in its subcellular distribution (Figure 12B,D).

## 3. Discussion

Observations from the current study clearly establish that the Cu status in the SVZ, the largest germinal region in adult brains, plays a critical role in regulating proliferation, migration, and differentiation of adult NPSCs along the SVZ-RMS-OB axis. This conclusion is supported by several lines of evidence. First, our in vitro neurosphere studies revealed that increasing Cu levels in the culture medium significantly suppressed NPSC’s differentiation and migration, while chelating Cu by D-Pen greatly reduced cellular Cu burden and restored cells’ differentiation and migration. Second, in vivo ICV infusion of D-Pen in mice, either by short-term (7-day) or long-term (28-day) infusion, selectively reduced Cu concentrations in the SVZ; this reduction significantly upregulated the proliferation of NPSC in SVZ and facilitated neuroblast migration in RMS. Third, long-term low-dose D-Pen infusion increased new neuron production in OB, especially in the GCL, without depleting the NSPC pool in the SVZ, suggesting the impact of the Cu status on cell differentiation in OB which may affect the olfactory function. Fourth, treatment with D-Pen in both of our in vitro and in vivo studies significantly influenced the expression of regulatory proteins participating in NPSC’s differentiation and migration, as well as cellular Cu transport and storage along the SVZ-RMS-OB axis. Finally, our data also revealed that infusion of D-Pen to the lateral ventricles greatly increased the expressions of CTR1 in the choroid plexus, suggesting a potential contribution of the choroid plexus in regulating the Cu level in the SVZ. 

A well-characterized in vitro SVZ-derived neurosphere model is essential to study the SVZ adult neurogenesis. Existing protocols have described workflows to isolate adult NSPCs [34,35]; however, these protocols allow only for simple qualitative assessment, but not for quantitative analyses of neuronal lineage differentiation and migration. The workflow established in this report fills these gaps in the following two critical steps. First, we adapted Leibovitz’s L-15 medium for freshly extracted brain in subsequent microdissection of SVZ. Compared to simple inorganic salts-based PBS and HBSS, L-15 medium contains amino acids, vitamins, and inorganic salts, which prevent the cell death of NSPCs upon the immediate isolation from the live brain tissues and ultimately facilitate the neurosphere formation. Second, poly-L-ornithine (PLO) has been used in the literature, often in combination with laminin, to promote the survival, migration, and differentiation of neurospheres derived from embryonic brain through ERK pathway [36,37]. Our protocol used PLO but replaced laminin with a polymer-made bottom to improve cell attachment. By comparing the migration area and final neuron production, it was evident that this approach greatly improved cell growth, even in comparison with embryonic neurospheres [36,37,38]. These improvements yielded nearly uniform neurospheres with an average 216.8 ± 5.4 µm (RSD% 2.5, n =10) in the diameter (Appendix A), giving the confidence in subsequent quantitative analyses of neurosphere dynamics between different treatments.

Applying D-Pen in our in vitro and in vivo experiments proved to be useful in reducing cellular and tissue Cu levels. Experiments with Phen Green, a dye specific to identifying the labile Cu pool which is different from the protein-bound Cu pool [39], indicated a reduced level of bioactive Cu in neurospheres following D-Pen treatment. Although D-Pen treatment by itself cannot completely remove Cu from the culture medium, the depleted intracellular labile Cu pool by D-Pen apparently effectively modulated adult neurogenesis by increasing neurosphere differentiation and migration (Figure 3A), and in cases of excess Cu in culture medium, it reversed Cu’s suppression of adult neurogenesis and Cu’s modulation of pertinent NSC regulatory factors (Figure 4 and Figure 5). Our results are consistent with the literature reports that only the labile, bioactive Cu ions mediate the Cu-related cellular activities [40]. 

Our recent work has established that systematic Cu disorders in adult rats can alter SVZ neurogenesis, leading to neurochemical imbalance in the SVZ-RMS-OB axis [24]. The Cu concentrations along the SVZ-RMS-OB axis in that study change significantly in both Cu-deficient and Cu-overload animals. However, the possibility of an overall systemic Cu dyshomeostasis on the SVZ-RMS-OB axis cannot be ruled out. Hence, this study adopted the ICV infusion approach by locally infusing D-Pen directly into the CSF in brain ventricle, making it possible to investigate the impact of CSF Cu homeostasis on the SVZ adult neurogenesis, under the assumption that D-Pen would selectively reduce Cu concentrations in SVZ, a ventricular region bathing in the CSF. Our AAS data showed that the ICV infusion of D-Pen indeed dose-dependently reduced Cu concentrations in the SVZ; but it did not affect other tested brain regions such as RMS, OB, hippocampus, and frontal cortex, except for striatum where Cu levels were increased. Noticeably, the low-dose D-Pen resulted in a marginal reduction of SVZ Cu (13.1%), which was not statistically significant (*p* = 0.19), as compared to a significant reduction in the high dose D-Pen infusion (21.4%). This marginal reduction could be due to the increased Cu level in striatum, an adjacent region to the SVZ. In fact, the SVZ is a thin layer covering striatum; it was possible that the process in isolating SVZ tissue may carry minor account of striatal tissue. Since D-Pen ICV infusion caused an elevated Cu in striatum, a minor amount of contaminated striatal tissue could confound the results of SVZ Cu, which would have shown a greater Cu reduction after low-dose D-Pen infusion. Importantly, the fact that no Cu level changes were found in other tested brain regions including RMS and OB suggests a local action of D-Pen directly on SVZ, which influenced adult neurogenesis in the SVZ-RMS-OB axis. 

One cycle SVZ adult neurogenesis takes approximately 28 days in mice [41]. Under this timeframe, the neuroblasts differentiated from SVZ NSPCs migrate in the RMS and subsequently reach OB, where they ultimately mature into two types of interneurons in two sub-regions of OB, i.e., PGCs in GL and GCs in the GCL. Our characterization of the SVZ-RMS-OB axis suggested that the low-dose D-Pen, but not the high-dose, facilitated the neurogenesis along the SVZ-RMS-OB axis. It was evident that following the short-term, 7-day infusion, more Ki67(+)/Nestin(+) cells emerged in the SVZ in the D-Pen (low) group. The observation was in line with the findings in RMS and OB, i.e., more BrdU(+)/DCX(+) neuroblasts observed in the OB following short-term, low-dose D-Pen infusion. However, for short-term, high-dose D-Pen treatment, more of such cells were found in RMS with fewer detected in the OB, implying that the high-dose D-Pen infusion, by yet undefined mechanism(s), arrested newly differentiated neuroblast cells in RMS. Similarly, in the long-term 28-day infusion, the D-Pen at low dose, but not the high dose, significantly increased newly generated BrdU(+)/NeuN(+) mature neurons in OB, especially in GCL. These findings suggest a dose-specific effect of D-Pen chelation of Cu in the CSF and/or SVZ on altering neurogenesis in the SVZ-RMS-OB axis, and only the low-dose D-Pen activates the SVZ neurogenesis. 

Stimulating the generation of adult NSPCs is not uncommon; but an aberrant activation often leads to a depleted NSPC pool in SVZ [42,43,44]. By examining the ratio of activated NSCs (aNSC) over the quiescent NSCs (qNSC), we found the low-dose D-Pen infusion increased the NSPC activation without eliciting notable alterations in the total NSC numbers, suggesting that the low-dose D-Pen (with a mild Cu reduction) did not deplete the NSPC pool. However, challenging the SVZ with the high-dose D-Pen seemed likely to deplete the NSPC pool, as evidenced by fewer GFAP(+)/CD133(+)-stained NSCs along the lateral ventricular wall. Taking together the overall dynamics of NSPCs in SVZ, RMS and OB, a mild reduction of Cu in the SVZ by the low-dose D-Pen treatment appeared to significantly increase newborn neurons in the OB, yet not at the cost of depleting the NSPC pool in the origin of SVZ. Since a depleted NSC pool, no matter partially or fully, can repopulate over a period of time [45,46], it would be interesting to investigate whether and how a prolonged D-Pen chelation may alter the NSC pool in the SVZ in our future experiments. 

Our data also showed the cluster-like enrichment of CTR1 and MT3, two critical proteins in regulating cellular Cu homeostasis, along the lateral ventricular wall in the SVZ, and their expressions were altered after D-Pen Cu chelation. These observations are in a good agreement with reports by our group and others [17,22,33]. Nonetheless, the current study revealed that ICV D-Pen infusion, while not changing the average CTR1 fluorescent intensity in the SVZ clusters, caused a translocation of CTR1 clusters from the cytosol toward the cell membrane facing the CSF. As CTR1 possesses the high capacity in taking up the extracellular Cu [47], this observation may reflect a higher demand of SVZ for Cu present in the CSF upon Cu chelation by D-Pen. In addition, quantitative analyses of CTR1 in striatum revealed that, upon Cu chelation, striatal CTR1 was increased in a dose-dependent manner (Figure 9), which may account for increased Cu levels following D-Pen treatment as shown by the AAS data.

Data in this study also showed an upregulated MT3 expression in the SVZ following ICV D-Pen infusion in a dose-dependent manner, which is consistent with the result from the systemic Cu-deficiency model by this lab [24]. This may indicate a compensatory mechanism by which the Cu-depleted cells demanded more Cu for intracellular storage. In addition, unlike other MT isoforms, MT3 reportedly acts as a growth inhibitory factor (GIF) in astrocytes, and a decreased astroglia GIF level was associated with neuronal loss in AD brains [48]. Thus, an increased MT3 may act as the neuronal modulatory GIF to modulate the SVZ cells. This hypothesis, however, needs further testing. 

Interestingly, our IHC data demonstrated that neither CTR1 nor MT3 clusters completely overlapped with GFAP and/or Nestin. This suggests that certain cell types not studied in this report may occupy the CTR1 and MT3; more importantly these yet-to-be-defined cell types may also greatly contribute to Cu enrichment in SVZ. Given the cluster-like distribution pattern of CTR1 and MT3, we hypothesize that other Cu-related proteins, including but not limited to DMT1, ATP7A, and ATP7B, may also follow this expression pattern in SVZ. We believe that the single-cell RNA-seq technique can be used in our future studies to identify the cell types of interest and to solve this “cluster” puzzle. 

The choroid plexus plays a critical role in regulating Cu homeostasis in the CSF [49,50,51]. The current finding on a significant increase of CTR1 in the choroid plexus after D-Pen chelation of Cu is in a good agreement with the literature reports by systematic Cu deficiency induced by restricting dietary Cu [24,52]. Given the anatomical closeness of the choroid plexus to the SVZ in brain ventricle and the fluid communication between the two via the CSF, it is reasonable to postulate that changes in Cu-regulatory functions in the choroid plexus may modulate adult neurogenesis in nearby SVZ. Again, this hypothesis deserves extensive experimental testing. 

How then does Cu regulate the adult neurogenesis in the SVZ? Extensive reports in the literature have linked Cu to neuronal differentiation, synaptic signal transduction, and neurodegeneration [24,53,54,55,56]. However, a role of Cu in neurogenesis, no matter in early postnatal phase or in adult brain, remains elusive; even less is known about the molecular regulatory mechanism taking place in the SVZ-RMS-OB axis. Several transcription factors and molecules are known to stimulate the NSPC proliferation (e.g., b-catenin, Notch1, and Shh), facilitate the differentiation of NSPCs to neuroblasts (e.g., Dlx-2), and guide the direction of newborn neuroblast migration (e.g., SLIT1, SLIT2, ROBO1, and ROBO2) [29,30,57]. The qPCR data from our in vitro neurosphere studies clearly showed that the expression of some of these critical transcription factors was sensitive to Cu levels in the surrounding environment. Moreover, the Cu-induced alteration in mRNA expressions of these neurogenesis regulators could be alleviated by a mild D-Pen chelation. Thus, we postulate that Cu may regulate the adult neurogenesis in SVZ by acting on these critical neurogenesis modulators. This assumption, however, does not exclude Cu interactions with other yet-to-be-identified factors that participate in SVZ adult neurogenesis, as Cu is a biologically highly active trace element. This hypothesis deserves further investigation. 

The other interesting observation from this study is that only a mild reduction of Cu by the low-dose D-Pen treatment activates neurogenesis in adult brain. Cu is an essential metal to brain cells, and yet any excess or deficiency of this metal causes neurotoxicity. Thus, an elegant balance is necessary in maintaining cellular Cu levels through Cu uptake, storage, and export (Zheng and Monnot, 2012). Interestingly, such processes were largely supported by organelles, especially Golgi apparatus and mitochondria [58,59,60]. Compared to SVZ neuroblast cells, NSPCs possess larger Golgi apparatus and more mitochondria [61]. Golgi apparatus facilitates intracellular Cu trafficking and exports excessive Cu through ATP7A and ATP7B [50,59] and mitochondria produce the energy required for NPSCs’ proliferation through the Cu-dependent tricarboxylic acid (TCA) cycle [62,63]. The low-dose D-Pen treatment may remove the Cu-related suppression of cellular modulators in adult neurogenesis, but not lead to a Cu deficiency-associated cytotoxicity, so that a mild Cu chelation may activate the neurogenesis process. In contrast, the high-dose D-Pen chelation of Cu may create a condition similar to Cu deficiency, which subsequently disrupts normal functions of mitochondria in NPSCs [64]. Interestingly, this assumption was supported by our finding that, when treated with higher concentrations of D-Pen, neurons differentiated from neurospheres exhibited extensive NeuN translocation from nucleus to cytosol, which was not elicited by D-Pen at 20 µM (Figure 3); the observation suggests that high-dose D-Pen caused cytotoxicity in NSPCs by disrupting the normal neural differentiation. Indeed, the reports from human Parkinson’s brain and animal PD models suggest that Cu deficiency is associated with PD pathology, impaired SOD1 and COX activities, and increased cellular oxidative stress [65,66,67,68]. Thus, it is possible that by over-chelation of intracellular labile Cu ions, the high-dose D-Pen treatment may generate the cytotoxicity that overshows its benefit. Taken together, Cu chelation in the SVZ can lead to increased adult neurogenesis; but the magnitude of the chelation needs to be well controlled.

## 4. Materials and Methods

### 4.1. Materials

Chemical reagents were purchased from the following sources: copper chloride dihydrate (MW 170.48 g/mol, purity > 99%), D-penicillamine (MW 149.21 g/mol, purity > 98%), 5-bromo-2′-deoxyuridine (Bromodeoxyuridine, BrdU), heparin sodium, and paraformaldehyde (PFA) from Sigma Aldrich (St. Louis, MO, USA); Nunclo Sphera 96-well, U-shaped-bottom microplate, wide bore pipette tips, neurobasal plus medium, B-27 plus supplement, GlutaMAX supplement, gentamicin (50 mg/mL), trypsin-EDTA (0.05%), defined trypsin inhibitor, DNAse I, epidermal growth factor (EGF), fibroblast growth factors (FGF), fetal bovine serum (FBS), Alexa Fluor 488-conjugated goat anti-rabbit IgG (H + L) (A-11008), Alexa Fluor 568-conjugated goat anti-chicken IgY (H + L) (A-11041), Cy5-conjugated goat anti-rat IgG (H + L) (A-10525), Phen Green (cell permeant) from Thermo Scientific (Waltham, MA, USA); Apoptosis assay kit from Abcam (Cambridge, MA, USA); normal goat serum (NGS) from Jackson ImmunoResearch (West Grove, PA, USA); 24 Well Plate with #1.5 glass-like polymer coverslip bottom from Cellvis (Mountain View, CA, USA); Cultrex ready-to-use poly-L-ornithine solution from R&D Systems (Minneapolis, MN, USA); and Triton X-100 from Bio-Rad (Hercules, CA, USA). All reagents were of analytical grade, HPLC grade, or the best available pharmaceutical grade.

Osmotic pumps (model 1007D, model 1004) and brain infusion kit 3 were purchased from DURECT Corporation (Cupertino, CA, USA); and Gluture topical tissue adhesive from MWI Animal Health (Boise, ID, USA). Information regarding primary antibodies, host species and dilution factors is provided in Appendix A.

### 4.2. Animals

C57BL/6 mice aged 3 months old were purchased from Envigo Inc. (Indianapolis, IN). Upon arrival, animals were housed in a temperature-controlled room under a 12 h-light/12 h-dark cycle and allowed to acclimate for one week prior to experimentation. Animals had free access to distilled-deionized water and Purina semi-purified rodent chow (Purina Mills TestDiet, Richmond, IN, USA) ad libitum. The study was conducted in compliance with standard animal use practices and approved by the Animal Care and Use Committee of Purdue University (PACUC No. 1112000526). 

### 4.3. Experimental Design

The Experiment 1 was to develop and characterize an in vitro neurosphere model for investigation of SVZ adult neurogenesis. This model allowed us to test D-Pen’s efficacy in mitigating Cu-induced neurosphere disruption as well as the underlying mechanisms. 

The Experiment 2 extended our in vitro studies to in vivo, to establish an ICV infusion animal model to explore how D-Pen administration locally by ICV infusion altered the brain Cu levels and to observe the ensuing adult neurogenesis along the SVZ-RMS-OB axis in response to short- and long-term D-Pen infusion at two doses. Within the long-term infusion timeframe, we also investigated the changes of the homeostasis of NSPC pool, and the expression patterns of CTR1 and MT3 by IHC, two critical Cu regulatory proteins, in the SVZ. 

Finally, the Experiment 3 extended our in vitro and in vivo studies on SVZ to investigate whether D-Pen infusion in vivo altered the expressions of Cu regulatory proteins (i.e., CTR1 and MT3) in the choroid plexus (CP), a blood-rich tissue in brain ventricles nearby the SVZ and regulating Cu transport in the CSF [49]. 

### 4.4. Establishment of an In Vitro Neurosphere Model for Adult Neurogenesis Studies

To establish an in vitro model of neurospheres for Experiment 1, three types of culture medium were prepared as follows. First, the neurosphere background medium (Medium-1) was prepared by mixing 48.5 mL Neurobasal Plus Medium with 1 mL B-27 Plus supplement, 0.5 mL GlutaMAX, and 50 μL Gentamicin (50 mg/mL). Of note, B-27 Plus supplement consists of a cocktail of antioxidants, which creates a reducing environment to improve adult NSPCs’ viability and their long-term survival. Second, the growth factor-enriched neurosphere induction medium (Medium-2) was prepared based on the Medium-1 by adding 2 μL EGF (0.1 mg/mL), 2 μL FGF (0.1 mg/mL), and 2 μL heparin stock solutions (10 mg/mL) per 10 mL Medium-1. Finally, the neurosphere differentiation medium (Medium-3) was prepared by adding 0.1 mL FBS into 10 mL of Medium-1 without the growth factor. 

A 3-month-old mouse was anesthetized by intraperitoneal (ip.) injection of ketamine (75 mg/kg) and xylazine (10 mg/kg). The whole brain was extracted and washed in ice-cold Leibovitz’s L-15 medium to remove excessive blood. The brain was then transferred to a 100 mm culture dish pre-filled with 10 mL ice-cold Leibovitz’s L-15 medium for microdissection. With the brain’s ventral aspect facing upward, a coronal cut was made by a razor blade at the optical chiasm illustrated by the dash line in Figure 1A. To dissect and collect the SVZ fraction, the rostral portion was placed against the dish with the coronal cut facing upward. Following removal of the septum, the Dumont curved #5/45 forceps was used to isolate the SVZ, a thin layer of tissue lining the lateral ventricle marked by the dash circle in Figure 1B. 

The isolated SVZ fraction was physically dissociated by mincing them gently with a scalpel blade for 1 min until no large tissues remained. The gentle mincing was necessary to avoid damaging the petri dish’s plastic material. The minced tissue was resuspended in 2 mL pre-warmed 0.05% Trypsin-EDTA and transferred to a 15-mL centrifuge tube. The preparation was incubated in a water bath at 37 °C for 7 min. Since the tissues tended to sink to the bottom of the tube, the tube was tapped every 1~2 min to facilitate the digestion during the incubation. This was followed by adding 2 mL of Defined Trypsin Inhibitor containing DNAse I at 0.01 mg/mL to terminate the digestion. After centrifugation at 300× *g* for 5 min, the pellet was resuspended in 1 mL of the neurobasal background medium and gently triturated with a 1-mL pipette tip up and down for 7 times. By adding another 4 mL of the Medium-1, the suspension was filtered through a 40-μm sterile cell strainer to remove the undissociated tissue chunks. This filtrate containing single cells or small aggregates was then centrifuged at 300× *g* for 5 min, followed by resuspending the pellet in 12 mL of the inductive Medium-2. The prepared suspension was seeded in one 12-well culture dish, with 1 mL suspension per well. The culture was maintained under 37 °C with 5% CO_2_ for 6 days.

The formation of neurospheres was recorded by tracking their daily growth (Figure 1C). Small neurospheres (diameter ~20 μm) were first observed on Day 3; larger neurospheres gradually formed in the days followed. On Day 6, neurospheres with desirable sizes (diameter 100~200 μm) became visible and afloat in the medium. During the preliminary experiments, it was noticed that extending the culture duration beyond 6 days caused the attachment of neurospheres to the bottom of the dish with ensuing spontaneous differentiation or the core of neurospheres turning dark. Therefore, the optimal culture duration to acquire high-quality primary neurospheres was 6 days.

On Day 6, the primary neurospheres were gently aspirated without disturbing the adherent debris through wide bore pipette tips to a 15 mL centrifuge tube. Following centrifugation at 300× *g* for 5 min, primary neurospheres were collected and then incubated with 0.05% Trypsin-EDTA for 2 min in a water bath at 37 °C in order to dissociate them into single cells; this process was necessary because the size of the original primary neurospheres varied widely, and the debris present in the culture adversely affected cell’s differentiation (Figure 1C). The dissociated cells were reseeded at 2.5 × 10^4^ cells/mL in a Nunclon™ Sphera™ 96-well U-shaped-bottom microplate (100 µL cell suspension per well). Following 2-day culture in Medium-2, a stable and robust neurosphere was formed in each dish, with an average diameter of 216.8 ± 5.37 µm (n = 10) (Figure 1D and Appendix A). These premium neurospheres were ready for subsequent characterization studies and other experimentation. 

### 4.5. Characterization of Neurospheres by Immunofluorescence

To characterize the proliferation of the formed neurospheres in Figure 1D, BrdU was added to the culture medium at a final concentration of 10 µM and the incubation was continued for 3 h. The neurospheres were gently aspirated, washed twice with PBS, and fixed in 4% PFA in PBS for 10 min at room temperature, followed by 2× PBS washes. Neurospheres were then incubated in 2 N HCl for 30 min under 37 °C to hydrolyze the DNA. The mixture was neutralized in 0.1 M sodium borate buffer (pH 8.5) for 30 min at room temperature. After two washes with PBS, samples were blocked and permeabilized in PBST containing 5% normal goat serum and 0.3% TritonX-100. The preparation was further incubated with primary antibodies against BrdU, Nestin, DCX, and/or NeuN at 4 °C overnight. Following 3 washes with PBST, neurospheres were incubated with fluorophore-conjugated secondary antibodies (1:500) at room temperature for 1 h protected from light. After another 3 PBST washes, neurospheres were mounted on microscope slides with mounting medium and coverslips for direct characterization by fluorescent microscopy. 

Another set of fresh neurospheres (Figure 1D) were then seeded on a PLO-coated surface in the differentiation Medium-3 to allow for attachment, migration, and finally differentiation into mature neurons. After 7 days in culture, the differentiated neurospheres were characterized by immunofluorescence as described above and using primary antibodies against NeuN and Nestin. 

### 4.6. Quantification of Apoptosis in Formed Neurospheres 

An apoptosis detection kit was used to examine whether neurospheres, formed under the current protocol, had a low level of apoptosis. Briefly, differentiated neurospheres were washed twice with the assay buffer provided by the kit, followed by incubation in the assay buffer containing Apopxin Green Indicator (Ex/Em = 490/525 nm) at room temperature for 1 h. After two washes with the assay buffer, images were captured under a fluorescent microscope for green apoptotic signals. The differentiated neurospheres treated with hydrogen peroxide at 1.6 mM for 30 min were used as the positive control for apoptotic signals.

### 4.7. Influence of Cu on Neurosphere Dynamics and D-Pen Intervention

To investigate how Cu exposure affected SVZ adult neurogenesis in vitro, neurospheres were cultured in the differentiation Medium-3 containing the final Cu concentrations at 1.0, 10.0, and 100.0 μM. The Cu treatment was terminated on day 7, and neurospheres were characterized by measuring the migration area and counting NeuN(+) mature neurons per neurosphere. For D-Pen intervention studies, the Medium 3 containing D-Pen alone at 20, 50, and 100 µM was used to culture neurospheres for 7 days, followed by assessment of neurosphere growth. From the above dose-response studies, the ideal concentrations for Cu and D-Pen were selected to investigate whether the presence of D-Pen in culture medium alleviated Cu-induced suppression in neurosphere growth. In these experiments, neurospheres grew in culture medium with (10 µM Cu) or without Cu overnight, followed by incubation with the medium supplemented with 20 µM D-Pen.

### 4.8. Estimation of Intracellular Cu Levels by Phen Green

Phen Green is a widely used fluorescent Cu sensor to estimate intracellular Cu levels in live cells. To determine intracellular Cu levels, the solid Phen Green (1 mg) was dissolved in 1 mL chloroform; the solution was aliquoted to separate vials and then the chloroform was allowed to evaporate to complete dryness in a light-proof fume hood. The dried vials with solid Phen Green were sealed and stored at −20 °C. Upon use, the chemical was resolubilized with 10 µL DMSO for live cell Cu imaging. Of note, the intensity of the Phen Green fluorescence correlates inversely to intracellular Cu levels due to Cu’s ability to quench its green fluorescence. In other words, neurospheres with a higher intracellular Cu level would emit less green fluorescence.

To quantify intracellular Cu status, neurospheres were cultured with the control medium, Cu (10 µM) medium, or Cu (10 µM) plus D-Pen (20 µM) medium for 7 days. These preparations were washed with PBS three times to completely remove the medium components. A serum-free background medium containing 5 µM Phen Green was added to neurospheres and incubation continued at 37 °C for 30 min. The Phen Green-loaded neurospheres were washed with PBS three times to remove unbound Phen Green; the samples were imaged under Nikon Confocal Microscope C1 (Tokyo, Japan). Average fluorescent intensity of a neurosphere was quantified by ImageJ. The intracellular Cu levels in neurospheres were estimated by the equation of (F_ct_ − F)/F_ct_, where F_ct_ and F represent the average fluorescence intensity of control neurospheres and experimental neurospheres, respectively. 

### 4.9. Quantitative Polymerase Chain Reaction (qPCR)

Expression of mRNAs encoding key proteins in regulating Cu homeostasis and SVZ adult neurogenesis were quantified in neurospheres by qPCR. Following the 7-day culture in the control medium, Cu (10 µM) medium or Cu (10 µM) plus D-Pen (20 µM) medium, total RNA was isolated from neurospheres using TRIzol Reagent as per manufacturer’s instructions. An aliquot of RNA (0.5 µg) was reverse-transcribed into cDNA using the BioRad iScript cDNA synthesis kit. After addition of specific primers, iTaq Universal SYBR Green Supermix was used to quantify the fluorescence during the amplifications. The qPCR program was run by the CFX Connect Real-Time PCR Detection System (Bio-Rad, Hercules, CA, USA) with an initial 3 min denaturation at 95 °C, followed by 40 cycles of 30 s denaturation at 95 °C, 10 s gradient 55.0–65.0 °C, and a 30 s extension at 72 °C. Dissociation curves were examined to verify that the majority of detected fluorescence was derived from the labeling of specific PCR products. Each qPCR reaction was run in duplicate. Relative mRNA expressions were calculated using 2^−ΔΔCt^ method with beta-actin as the reference gene. The forward and reverse primers for genes of interest in this study were designed by Primer Express 3.0 software and listed in Appendix A.

### 4.10. Intracerebroventricular (ICV) Infusion of D-Pen by Alzet Osmotic Pumps

The ICV infusion used in Experiment 2 was graphically illustrated in Figure 6A. Briefly, osmotic pumps were loaded with D-Pen dissolved in saline as per manufacturer’s instructions; pumps loaded with saline were used as controls. Specifically, the pump model 1007D and model 1004 were manufactured to infuse for 7 days at 0.5 µL/h and for 28 days at 0.11 µL/h, respectively. The D-Pen solution was prepared to administer at low (0.075 μg D-Pen/h) or high (0.75 μg D-Pen/h) dose during the two above infusion durations. The loaded pumps were connected to the Alzet brain infusion kit 3 to establish a pump-tubing-cannulation system. Prior to surgical cannulation, the pumps were primed by incubating in saline under 37 °C overnight (pump model 1007D) or 48 h (pump model 1004), after which the primed pumps were surgically implanted as follows.

Mice that had received three daily doses of BrdU at an interval of approximately 24 h (50 mg/kg, ip.) were individually placed in an anesthesia-induction chamber with the oxygen flow rate adjusted at 0.3 L/min and the isoflurane concentration at 3%. Once fully anesthetized, verified by no response upon a toe pinch, the head was fixed onto a stereotaxic device (KOPF Model 1900) with consistent isoflurane flow through the nosecone. Eye ointment was applied onto each eye, and hair was shaved from the eye level down to top of shoulder. A midline incision was made from eye level down to the shoulder area. A curved hemostat was then slowly inserted into the back skin of the animal with the curved end faced up to create a subcutaneous space for pump insertion. The pump was then gently inserted into the subcutaneous space and the skin was closed. The cannula holder was stereotaxically adjusted above the bregma. The coordination for the ventricular insertion was at 1.0 mm left and 0.2 mm posterior to the bregma (lateral/medial: 1.0 mm; anterior/posterior: −0.2 mm). The cannula’s stainless-steel tube was allowed to penetrate the skull (2.5 mm in depth) and tightly glued onto the skull for 15 s. The wound was closed by applying Gluture topical tissue adhesive alongside the cut. Lidocaine hydrochloride jelly and triple antibiotics ointment were topically applied on the wound, followed by a subcutaneous administration of ketoprofen (5 mg/kg). The postoperative mouse was placed on a heat pad, and typically recovered within 15 min after surgery. The pump-implanted mice were housed separately, with their conditions tightly monitored throughout the ICV infusion.

### 4.11. Determination of Cu Concentrations by Atomic Absorption Spectrometry (AAS)

Upon completion of ICV infusions of D-Pen or saline, mice were anesthetized and transcardially perfused with 20 mL ice-cold PBS and 20 mL ice-cold 4% PFA in PBS sequentially. A successful perfusion was confirmed by a tremor upon PFA perfusion. One half of the brain was used for IHC studies as described below. The other half of the brain was dissected for AAS analysis of Cu levels. Brain regions for AAS included SVZ, RMS, OB, hippocampus (HP), striatum (ST), and frontal cortex (FC). The SVZ tissue that covers the surface of the lateral ventricle was carefully cut off by a microdissection spring scissors under a microscope. Brain tissues were digested by ultrapure nitric acid in the MARSX press microwave-accelerated reaction system. Agilent Technologies 200 Series SpectrAA with GTA 120 graphite tube atomizer was used to quantify Cu concentrations. Digested samples were diluted properly to keep each reading within the linear range (0–50 µg/L). The detection limit of Cu by this AAS assay was 0.9 ng/mL. The intra-day and inter-day precisions of the method for Cu were 1.6% and 3.7%, respectively [69]. 

### 4.12. Brain Slice Preparation and Immunohistochemistry (IHC)

The PFA-perfused half brain was further fixed in 20 mL 4% PFA in PBS under 4 °C overnight with slight agitation. Following 3× PBS washes, the fixed brain was dehydrated in 30% sucrose solution in PBS under 4 °C. Brains were expected to sink within 72 h. The dehydrated brains were then coronally or sagittally cut by a microtome into 40-µm brain slices. Specifically, coronal slices were serially placed in a 12-well plate to study changes in SVZ, OB, or RMS in the coronal view; while sagittal slices were serially harvested into a 6-well plate to investigate RMS changes. Each well accounted for 1/12 or 1/6 of the total brain sections, respectively. The brain slices were preserved in cryopreservation medium (30% sucrose, 1% polyvinylpyrrolidone, 30% ethylene glycol in 0.1 M phosphate buffer) under −20 °C.

For IHC analyses, brain slices were picked up by a paintbrush and washed in PBS 3 times to remove sucrose. For samples for BrdU analyses, slices were incubated in 2 N HCl for 30 min under 37 °C to hydrolyze the DNA. The acid-treated slices were then neutralized by 0.1 M boric solution (pH 8.5) for 30 min. Following a PBS wash, these slices were blocked in PBST containing 5% normal goat serum and 0.3% TritonX-100 for 1 h at room temperature. Blocked slices were then incubated with primary antibodies overnight at 4 °C. Following 3× PBST washes, slices were then incubated with fluorophore-conjugated secondary antibodies (1:500) for 1 h at room temperature in the dark. After 3 PBST washes, slices were counterstained with DAPI and then mounted onto microscope slides with mounting medium and coverslips. The IHC images were captured by Nikon A1Rsi Confocal system. Z-stack scanning and large-image stitching were conducted for brain slice 3D reconstruction and the subsequent cell counting. Information about antibodies used in this section was provided in Appendix A.

### 4.13. Statistical Analysis

Data in this report were presented as mean ± standard deviation (SD). Statistical analyses of the differences among groups were carried out by one-way ANOVA with post hoc comparisons by the Dunnett’s test. Statistical column graphs were generated by GraphPad Prism 8 (San Diego, CA, USA), and all the statistical analyses were conducted using the embedded statistical programs. Differences were considered significant if *p* values were equal or less than 0.05.

## 5. Conclusions

By using optimized in vitro neurosphere model and in vivo ICV infusion technique, we have observed a distinct role of Cu in regulating adult neurogenesis along the SVZ-RMS-OB axis. The presence of a high level of Cu in the SVZ appears to be required for suppressing the initiation of adult neurogenesis; a moderate reduction of Cu levels in the SVZ by chelating labile Cu with Cu-chelator D-Pen can effectively activate adult neurogenesis in the SVZ, evidenced by increased NPSC proliferation in this niche origin, their migration via RMS, and ultimately their differentiation in OB. Importantly, our data provide the direct evidence that changes in Cu levels in neurospheres can change the expression of mRNAs encoding key proteins known to regulate SVZ adult neurogenesis. Moreover, distorted expressions of CTR1 and MT3, two critical Cu regulatory proteins in SVZ, may reflect the compensatory reaction of cells in response to altered Cu levels inside the cells or surrounding environment. Significant changes in CTR1 and MT3 expressions in the choroid plexus in response to D-Pen treatment suggest an indispensable role of this ventricular tissue in mediating SVZ neurogenesis in adult brain. Our findings are the beginning of our understanding of the regulatory mechanism by which Cu stabilizes the expression of NPSCs in SVZ and consequently influences the function of the SVZ-RMS-OB axis.

## Figures and Tables

**Figure 1 ijms-23-09888-f001:**
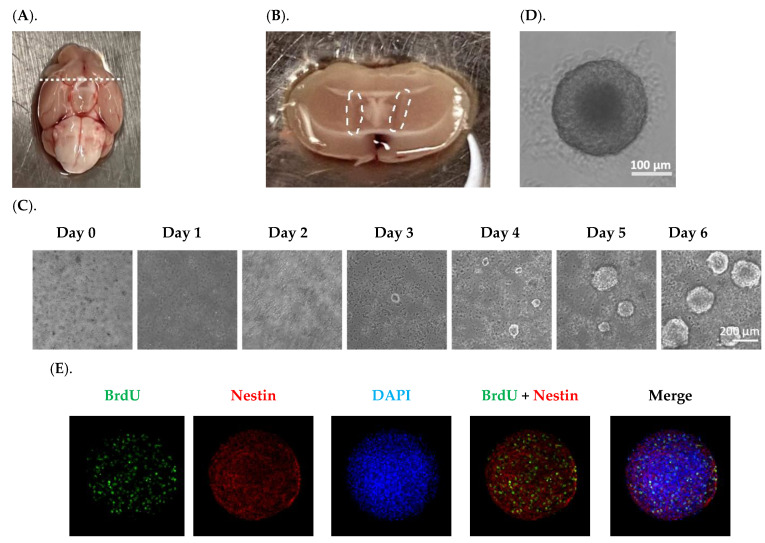
Establishment and characterization of in vitro neurospheres derived from adult SVZ. (**A**). An adult mouse brain was in a supine position; the dash line marks the optical chiasm where a coronal cut was made to expose the SVZ. (**B**). The SVZ tissues marked by the dash circle were isolated for disassociation. (**C**). The primary neurosphere formation was recorded for six days. Scale bar = 200 μm. (**D**). A representative image of a secondary neurosphere. Scale bar = 100 μm. (**E**–**G**). Characterization of secondary neurospheres by BrdU incorporation and staining with Nestin, DCX, and NeuN, respectively. Scale bar = 200 μm. (**H**). Characterization of a differentiated neurosphere by Nestin and NeuN. Center of the neurosphere was marked by the dash lines. Scale bar = 200 μm. (**I**). Neurospheres cultured in differentiation medium for 7 days showed negligible apoptotic signals, in contrast to neurospheres treated with 1.6 mM H_2_O_2_ for 30 min showing robust apoptotic signals in (**J**). Centers of the neurospheres were marked by the dash lines. Scale bar = 200 μm.

**Figure 2 ijms-23-09888-f002:**
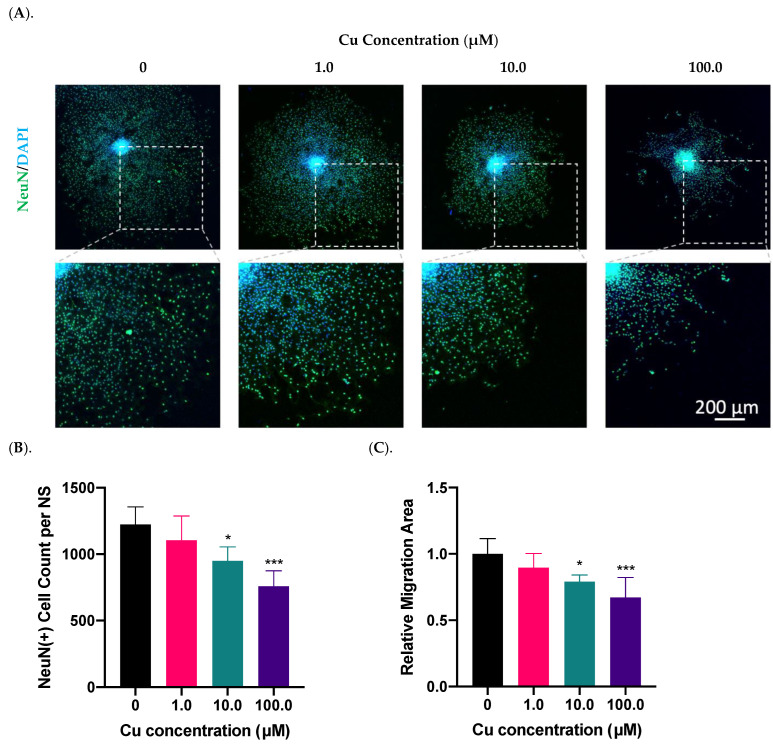
Cu dose-dependent suppression of neurosphere growth in vitro. (**A**). Differentiation and migration of SVZ neurospheres in response to Cu exposure in the culture medium at 1.0, 10.0, and 100.0 µM Cu. Differentiated mature neurons were labeled with NeuN. Scale bar = 200 μm. (**B**). NeuN(+) mature neurons per neurosphere (NS) were quantified in each group. Data represent mean ± SD, n = 6; * *p* < 0.05, *** *p* < 0.001 as compared with controls. (**C**). The relative migration area of neurospheres was quantified and normalized to the controls. Data represent mean ± SD, n = 6; * *p* < 0.05, *** *p* < 0.001 as compared with controls.

**Figure 3 ijms-23-09888-f003:**
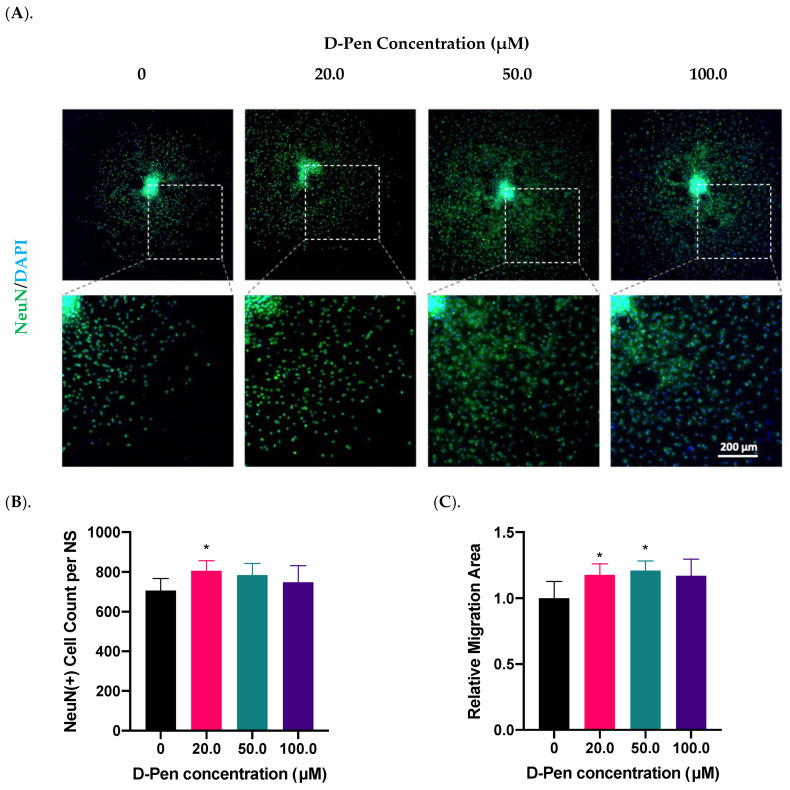
Treatment with D-Pen alone enhanced neurosphere growth in vitro. (**A**). D-Pen treatment exhibited dose-specific increases in neurosphere differentiation and migration. Neurospheres were stained with NeuN. Scale bar = 200 μm. (**B**). NeuN(+) mature neurons per neurosphere were quantified in each D-Pen treatment group. Data represent mean ± SD, n = 6; * *p* < 0.05 as compared with controls. (**C**). Migration areas of neurospheres were quantified and normalized to the controls. Data represent mean ± SD, n = 6; * *p* < 0.05 as compared with controls.

**Figure 4 ijms-23-09888-f004:**
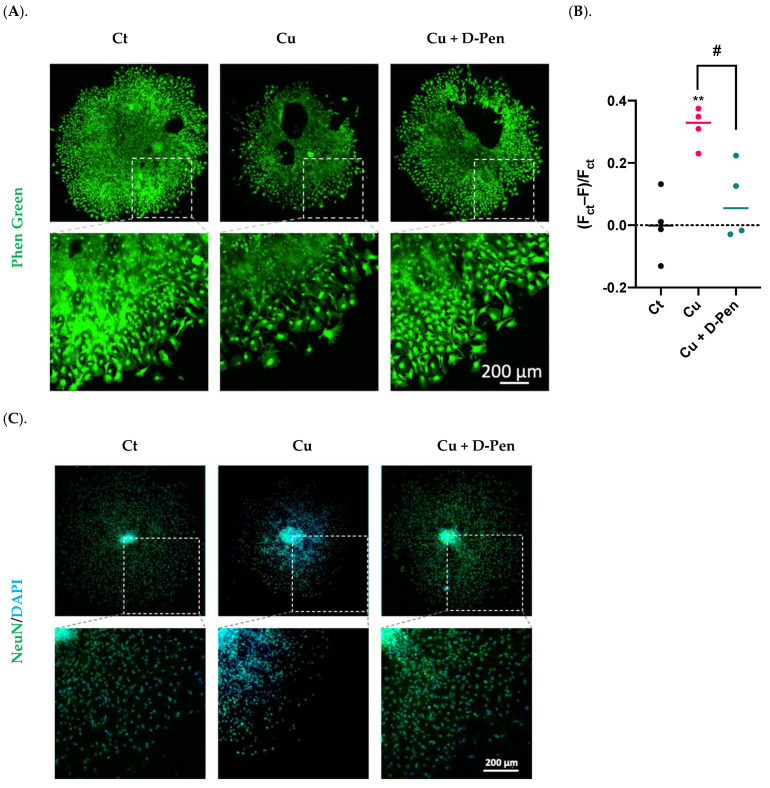
Chelating Cu by D-Pen restored Cu-suppressed neurosphere growth in vitro. (**A**). Phen Green imaging of differentiated neurospheres treated with 10-µM Cu alone or 10-µM Cu with D-Pen at 20 µM. Control (Ct) neurospheres were cultured in Medium 3. Scale bar = 200 μm. (**B**). Estimation of intracellular Cu burden by values of (F_ct_ − F)/F_ct_, where F_ct_ and F represents the average fluorescence intensity for controls and experimental neurospheres, respectively. Data represent mean ± SD, n = 4; ** *p* < 0.01 as compared with controls; # *p* < 0.05 as compared with Cu. (**C**). Neurosphere differentiation and migration as affected by Cu alone or by D-Pen chelation. Control (Ct) neurospheres were cultured in Medium 3. Neurospheres were stained with NeuN. Scale bar = 200 μm. (**D**). NeuN(+) mature neurons per neurosphere were quantified in each group. Data represent mean ± SD, n = 6; * *p* < 0.05 as compared with controls. (**E**). Migration areas of neurospheres were quantified and normalized to the controls. Data represent mean ± SD, n = 6; * *p* < 0.05 as compared with controls.

**Figure 5 ijms-23-09888-f005:**
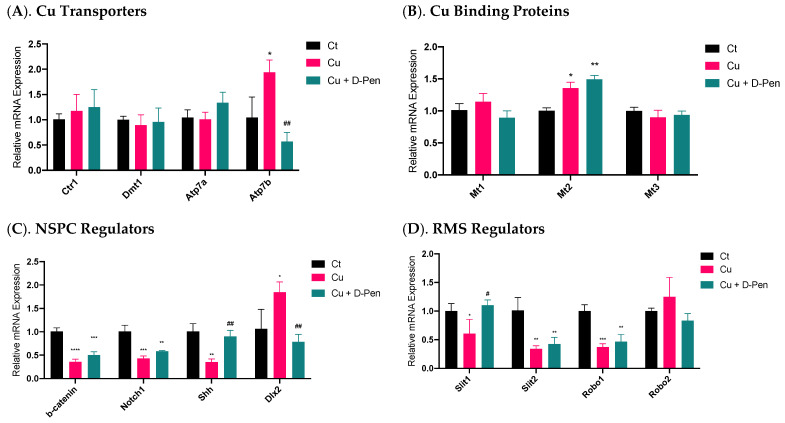
Expressions of mRNAs encoding critical modulators of Cu homeostasis and SVZ adult neurogenesis as affected by Cu status in culture medium. (**A**). Relative mRNA expressions of Cu transporting proteins in neurospheres, including Ctr1, Dmt1, Atp7a, and Atp7b. (**B**) Relative mRNA expressions of Cu binding proteins in neurospheres, including Mt1, Mt2, and Mt3. (**C**). Relative mRNA expressions of modulators of adult NSPCs in adult SVZ, including b-catenin, Notch1, Shh, and Dlx2. (**D**) Relative mRNA expressions of neuroblast migration modulators in adult RMS, including Slit1, Slit2, Robo1, and Robo2. For panel A-D, data represent mean ± SD, n = 3/group. * *p* < 0.05, ** *p* < 0.01, *** *p* < 0.001, **** *p* < 0.0001, as compared with the control group; # *p* < 0.05, ## *p* < 0.01, as compared with the Cu treatment group; 10 µM Cu and 20 µM D-Pen were used as previously demonstrated.

**Figure 6 ijms-23-09888-f006:**
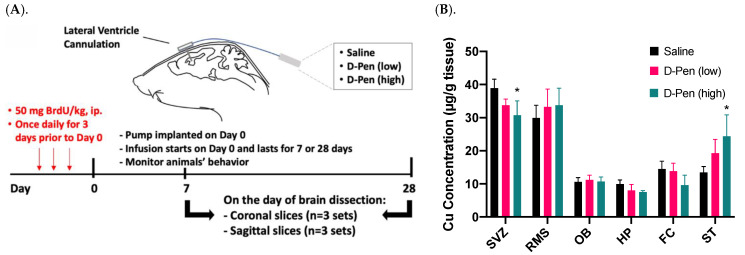
ICV experimental approach in mice and Cu concentrations in brain regions after short/long term D-Pen infusion. (**A**) A graphical illustration of the experimental design. Mice received ICV infusion of saline or D-Pen (0.075 μg/h or 0.75 μg/h) for 7 days (short term) or 28 days (long term). (**B**) Quantification of brain Cu concentrations by AAS in SVZ, RMS, OB, hippocampus (HP), frontal cortex (FC), and striatum (ST). Data represent mean ± SD, n = 3/group. * *p* < 0.05 as compared with the controls infused with saline.

**Figure 7 ijms-23-09888-f007:**
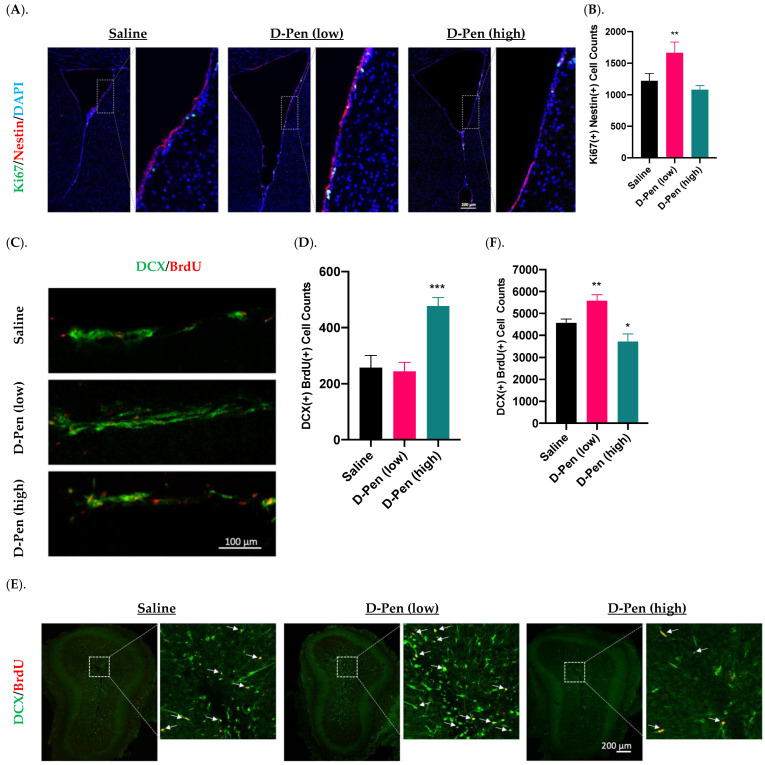
Adult neurogenesis in SVZ-RMS-OB axis following short-term ICV infusion of D-Pen. (**A**). Proliferation in the SVZ following short term ICV infusion of D-Pen. Proliferative NSPSs were Ki67(+)/Nestin(+). Scale bar = 200 μm. (**B**). Quantification of Ki67(+)/Nestin(+) cells per brain based on confocal 3D reconstruction. Data represent mean ± SD, n = 3/group; ** *p* < 0.01 as compared with the saline-infused controls. (**C**). DCX(+)/BrdU(+) newborn neuroblasts in the RMS from sagittal slices following short term ICV infusion of D-Pen. Scale bar = 100 μm. (**D**). Quantification of DCX(+)/BrdU(+) newborn neuroblasts per brain based on confocal 3D reconstruction. Data represent mean ± SD, n = 3/group; *** *p* < 0.001 as compared with the saline-infused controls. (**E**). Newborn neuroblast arrival at OB following short term D-Pen infusion. Newborn neuroblasts were DCX(+)/BrdU(+) marked by white arrows. Scale bar = 200 μm. (**F**). Quantification of DCX(+)/BrdU(+) newborn neuroblasts at OB per brain based on confocal 3D reconstruction. Data represent mean ± SD, n = 3/group. * *p* < 0.05, ** *p* < 0.01, as compared with the saline-infused controls.

**Figure 8 ijms-23-09888-f008:**
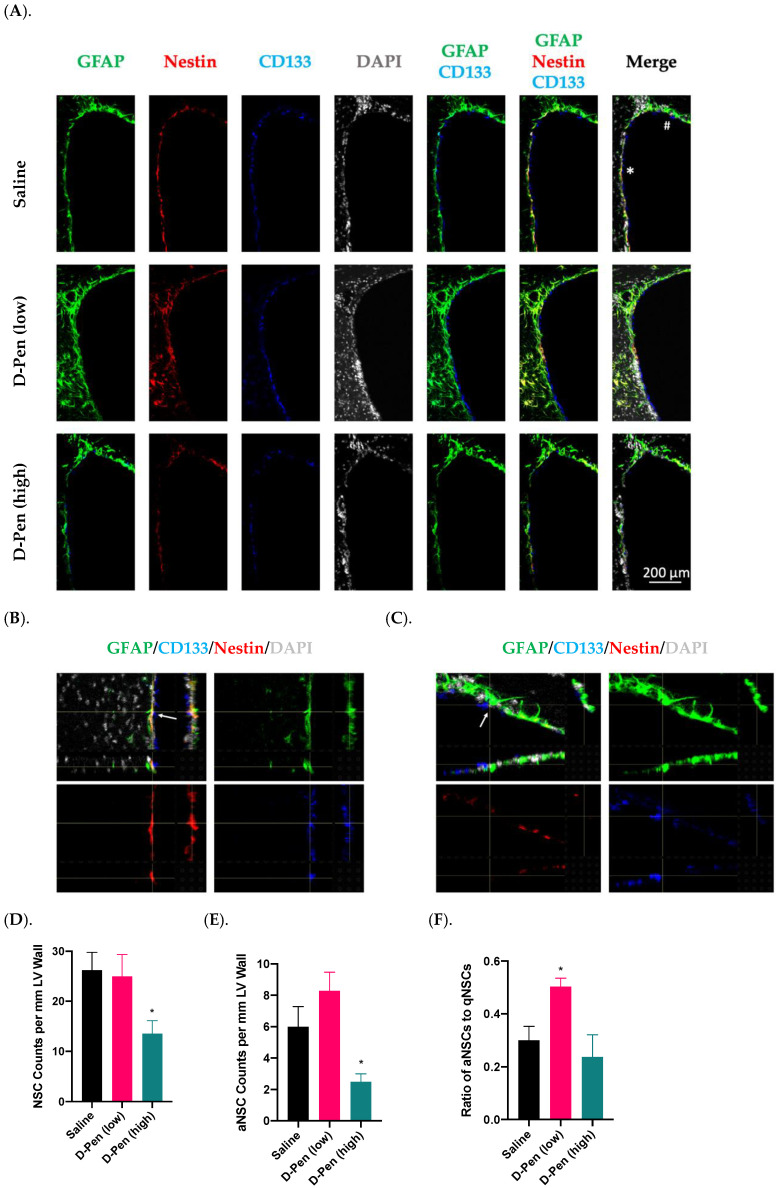
Adult NPSC pools in SVZ following long-term ICV D-Pen infusion. (**A**). Adult NSPC pool and cell identities revealed by staining with GFAP, CD133, and Nestin. Scale bar = 200 µm. (**B**). The asterisk-marked SVZ of a saline-infused mouse showing an aNSC: GFAP(+)-CD133(+)-Nestin(+). An arrow was used to show signal overlapping within this cell. (**C**). The pound-marked SVZ of a saline-infused mouse showing a representative qNSC: GFAP(+)-CD133(+)-Nestin(-). An arrow was used to show signal overlapping within this cell. (**D**). Quantification of GFAP(+)-CD133(+) NSCs in the anterior SVZ. (**E**). Quantification of GFAP(+)-CD133(+)-Nestin(+) aNSCs in the anterior SVZ. (**F**). Ratio of aNSCs to qNSCs in the anterior SVZ. For panels (**D**–**F**), data represent mean ± SD, n = 3/group. * *p* < 0.05, as compared with the saline-infused controls.

**Figure 9 ijms-23-09888-f009:**
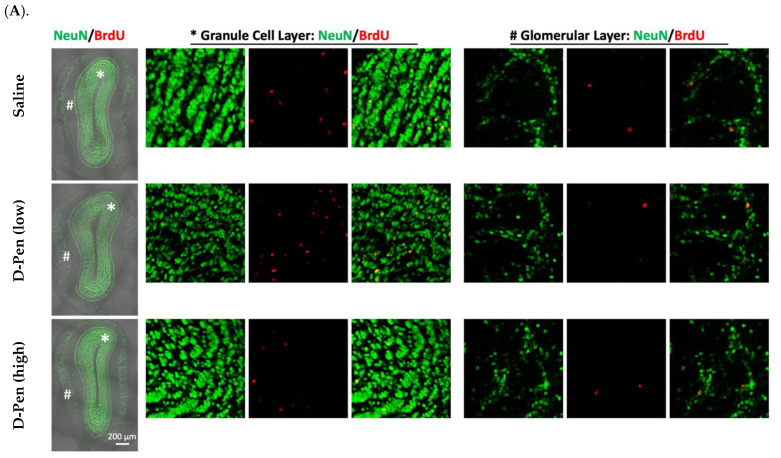
Adult neurogenesis in SVZ-RMS-OB axis following long-term ICV infusion of D-Pen. (**A**). Newly generated mature neurons in the OB’s GCL and GL labeled with NeuN and BrdU. Scale bar = 200 µm. The asterisk-marked area in GCL and pound-marked GL were further magnified and provided on the right, respectively. (**B**). Quantification of NeuN(+)/BrdU(+) newborn neurons in GCL and GL, and these two regions combined based on confocal 3D reconstruction. Data represent mean ± SD, n = 3/group. * *p* < 0.05, as compared with the controls.

**Figure 10 ijms-23-09888-f010:**
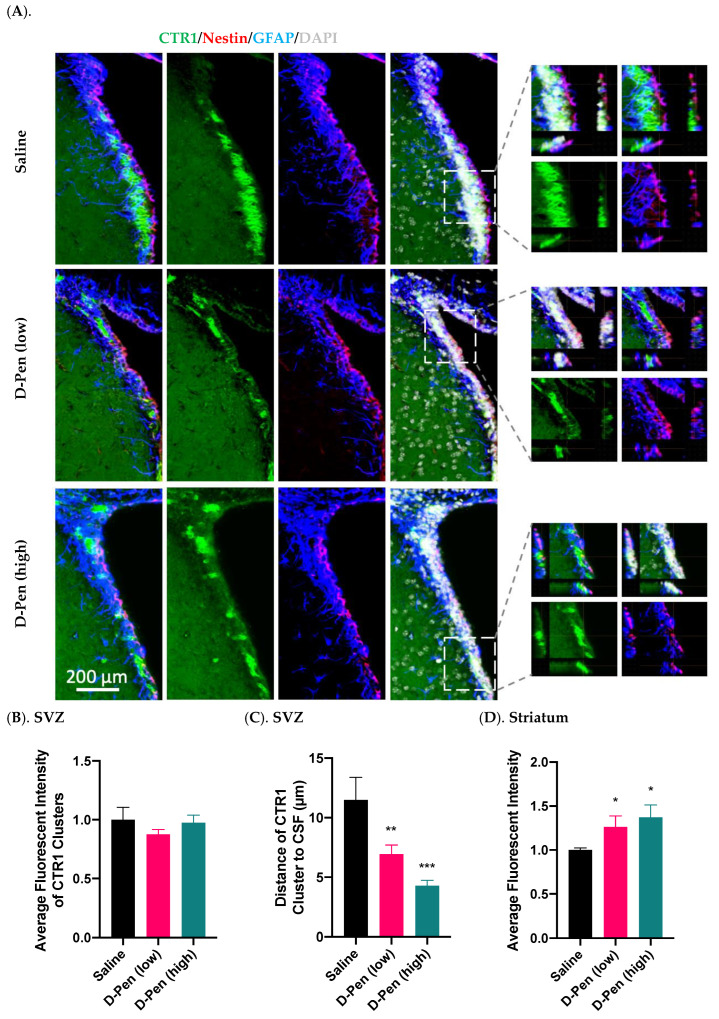
Expression of CTR1 in SVZ and striatum following long-term ICV D-Pen infusion. (**A**) CTR1 expression in the SVZ and colocalization with adult NSPCs. CTR1 clusters were further 3D-magnified for improved observations. Scale bar = 200 µm. (**B**) Quantification of average immunofluorescent intensities of CTR1 clusters in SVZ using multiple slices per brain. (**C**) Quantification of the average distance of CTR1 clusters in SVZ to the CSF using multiple slices per brain. (**D**) Quantification of average CTR1 immunofluorescent intensity in striatum using multiple slices per brain. For panels (**B**–**D**), data represent mean ± SD, n = 3/group. * *p* < 0.05, ** *p* < 0.01, *** *p* < 0.001, as compared with the controls.

**Figure 11 ijms-23-09888-f011:**
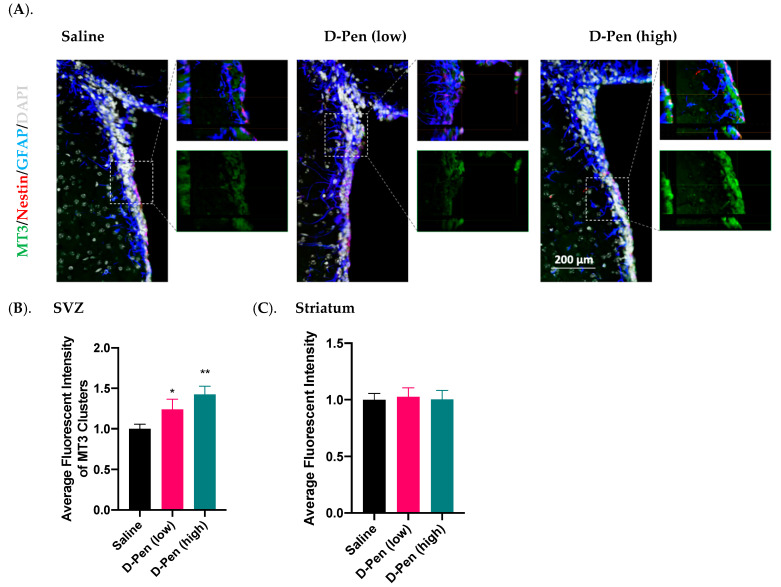
Expression of MT3 in SVZ and striatum following long-term ICV D-Pen infusion. (**A**) MT3 expression in the SVZ and colocalization with adult NSPCs. MT3 enrichment in the SVZ was further 3D-magnified for improved observations. Scale bar = 200 µm. (**B**) Quantification of MT3 immunofluorescent intensity in SVZ using multiple slices per brain. (**C**) Quantification of MT3 immunofluorescent intensity in striatum using multiple slices per brain. For panels (**B**,**C**), data represent mean ± SD, n = 3/group. * *p* < 0.05, ** *p* < 0.01, as compared with the controls.

**Figure 12 ijms-23-09888-f012:**
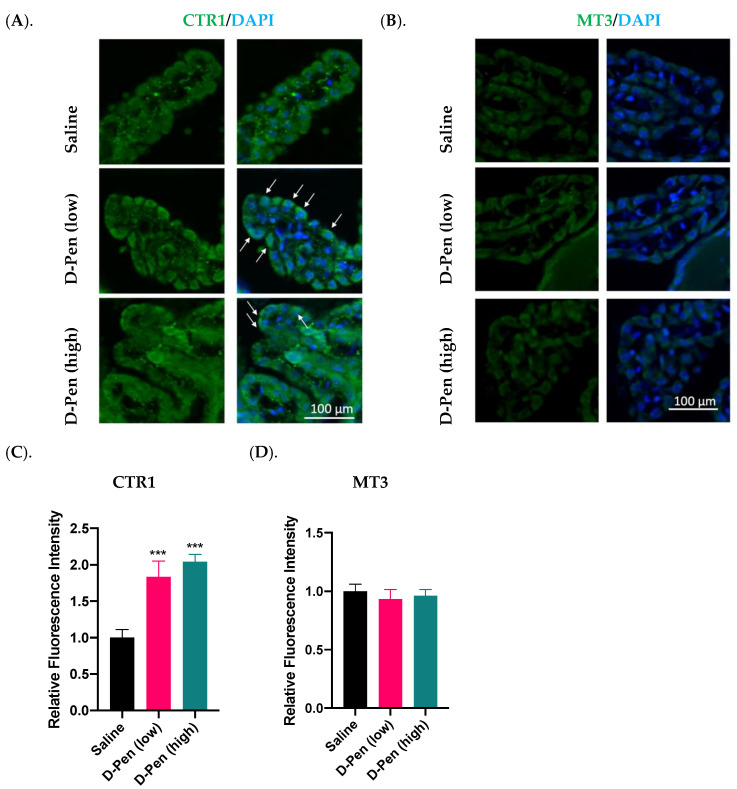
Expression of Cu regulatory proteins in the choroid plexus following long-term ICV infusion of D-Pen. (**A**). Expression of CTR1 in the choroid plexus. White arrows indicate CTR1 translocation toward the apical aspect facing the CSF. Scale bar = 100 µm. (**B**). Expression of MT3 in the choroid plexus. Scale bar = 100 µm. (**C**). Quantification of average CTR1 immunofluorescent intensity in choroid plexus using multiple slices per brain. (**D**). Quantification of average MT3 immunofluorescent intensity in choroid plexus using multiple slices per brain. For panels (**C**,**D**), data represent mean ± SD, n = 3/group. *** *p* < 0.001, as compared with the controls.

## Data Availability

The data presented in this study are available on request from the corresponding author.

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
