# Peer review of "Copper Modulates Adult Neurogenesis in Brain Subventricular Zone"

_ijms, 2022, doi:10.3390/ijms23179888_

Round 1
Reviewer 1 Report
The authors have written an extensive and quite interesting article on the role of copper in modulating adult neurogenesis in the SVZ using in vitro and in vivo methods.
However, it seems that the article contains too much information and it is easy to get lost in all the figures and results. The article should be more concise and the authors should focus on the main results that were well presented in the abstract. I suggested the transfer of some figures to the supplement and highlighted these comments in yellow: Figure 4, Figure 6B, Figure 12, Table 1, and Table 2. In addition, extensive changes and many modifications should be made for further consideration. Please answer all of these querries carefully and highlight all changes in color in the text.
ABSTRACT AND INTRODUCTION:
The summary is generally well started, but further in the methods section of the abstract is quite confusing. The methods should be more clearly stated.
1) Lines 21-24: Is this compared to saline-infused controls? If so, this should be mentioned here in the abstract to make it easier to follow.
2) Line25-26: Which key proteins?
3) Line27: what kind of investigation? Please be specific here!
4) Line27-29: It is not clear why the authors mention that MT3 changed, as there were no obvious or significant changes in the results (neither gene nor protein expression). This should be carefully described or presented since MT3 is one of the main targets of this study! If there were changes only in protein expression, than this should be only mentioned.
5) The sentences in line 25 and line 27 should be linked (connected), they do not need to be separated.
6) Line 28: CTR1 and MT3 need full word explanation. Is this protein expression or gene expression?
7) DCX should be explained as a whole word (doublecortin), here in the abstract and further in the introduction section where it is first mentioned, I suppose in line 106. It is not explained anywhere.
8) Line 72: CSF is not explained.
9) Line 72: "intracerebroventricular (ICV)" and line 95: "intracerebral ventricular (ICV): Please decide!
10) Line 99: why here the abbreviation "IHC"? why not in line 97? (i.e., immunohistochemical).
11) Line 99: gene expression for Cu regulatory proteins is missing.
12) here I have a question: why not PCR from tissue, not only from neurospheres?
METHODS PART:
1) Line 646 (or 4.3. Experimental Design ): If you already mentioned the protein expression of CTR1 and MT3 in experiment number 1, please mention the evaluation of the other expressions in experiment number 2.
2) Line 652: expression pattern of genes or proteins (MT3 and CTR1)?
3) Line 654: I suppose the experiment number 3 was performed in vivo like experiment number 2! In fact, Exp 3 is the same experiment as exp 2. If so, there is no need to call this experiment 3 if only different methods were used. Experiment 3 should have a completely different design. Please explain only the method and change it throughout the text. Experiment number 3 should simply be a continuation of the explanation of experiment number 2. Delete it.
4) Line 634: Table 1. please move to supplement (too much data).
5) Line 775: Fluorescence microscope: manufacturer, city, country are missing.
6) Line 794: please move Table 2. to the supplement (too much data).
RESULTS:
1) FIGURE 1: line 109: why no double staining DCX with Brdu to see continuity of differentiation from newborn neurons, i.e., neuroblasts to mature neurons? This is missing. In the other figure Neun marker is sufficient.
2) FIGURE 3: DAPI fluorescence is very low or if any. Maybe on the 3rd and 4th image only very slightly! Please change the representative images or increase the visibility. Also the green fluorescence should be more visible.
3) FIGURE 4: This image should be moved to a supplement. Too much data.
4) FIGURE 4: Please explain "Ct" in the figure legends (A,B,C,D,E).
5) FIGURE 4C: Again, the DAPI fluorescence has too low visibility. Please improve this.
6) FIGURE 6: Please insert the days for short- and long-term D-Pen infusion in the figure legend to make it easier to follow.
7) FIGURE 6: Why are the results extended to other parts of the brain? This is unnecessary. Please focus only on SVZ (along with RMS and OB). Figure 6B should be in the Supplement.
8) FIGURE 7: line 288: I guess the authors meant Fig7E instead of Fig7F?
9) FIGURE7C and 7E: too low visibility of red fluorescence (Brdu). This should be corrected.
10) FIGURE 8A: again low visibility of fluorescence for Nestin, CD133 and DAPI. Please improve this.
11) FIGURE 8B and 8C: lines 321-325: quote "An asterisk-marked aNSC and a pound-marked qNSC in a saline-infused SVZ (Fig. 8A, upper right panel) were further magnified in Figs. 8B and 8C, respectively, to reveal the identity of the cells. Of note, unlike GFAP and nestin, which are localized in the cytosol, CD133 was found "floating" in the CSF, as previously reported [32]." ...............From the images, it is very difficult to conclude anything. Please change the magnification method, i.e., edit them (e.g., in Photoshop or otherwise), and please do not leave them in this native form. They do not look good at all.
12) FIGURE 9: Again, improve the visibility of the red fluorescence.
13) FIGURE 10A: As mentioned in Figure 8, please edit the magnification images. Please do not leave them as they are in the original form.
14) Why CTR1 expression wasn’t presented in short-term D-Pen ICV infusion?
15) FIGURE 10B and 10D: Please change the labeling of these two graphs. They are not different ("relative fluorescence intensity"). Insert "average immunofluorescence of cluster..." or otherwise.
16) SVZ and ST abbreviations should be explained in all figure legends.
17) FIGURE 11: as mentioned in Figure 8 and Figure 10, please edit the magnification images. Please do not leave them as they are in the original form.
18) Why MT3 expression wasn't presented in short-term D-Pen ICV infusion?
19) FIGURE 11A: MT3 expressions look pretty much the same in all 3 magnifications. Please change the representative pictures or increase the visibility of green fluorescence.
20) FIGURE 11 B and C: As mentioned in Figure 10, change the labeling for these two graphs. They do not differ ("relative fluorescence intensity"). Inser of "SVZ" or "ST" or otherwise.
21) SVZ and ST abbs. should be explained in all figure legends
22) FIGURE 12: This figure should be transferred to the Supplement. Authors should focus on the main results. Too much data.
23) FIGURE 12: as mentioned in Figure 11, the CTR1 expressions look pretty much the same in all 3 figures. The graphs do not follow the figures. Please change the representative images or increase the visibility of the green fluorescence to avoid backgrounds.
Author Response
Thank you for your constructive comments and suggestions. We have addresses concerns in this resubmittal. Please see our point-to-point responses in attached file. -Wei Zheng

Reviewer 2 Report
In this manuscript, authors have used appropriate experiments to understand the copper role in neurogenesis. There are few modifications necessary to be addressed before this manuscript is published. Point-point comments are mentioned below.
1. In this manuscript authors have not given the procedure they have followed for administering BrdU in mice. They need to mention the time interval and route of administration need to be appropriately mentioned.
2. The figures containing NeuN/Dapi, where authors have used Green/Blue color is not clearly showing the difference between those dyes in the picture. Hence it is recommended to show the individual channels for easy understanding to the readers.
3. Similarly, in figure 7 A & E the individual channel images were not giving the information authors would want to explain.
4. I see that there is proper copper dose dependent effect observed in the in vitro experiments. However, in vivo experiments where D-Pen has been used there is clear opposite effect presented between the two different concentrations. Authors need to properly address this in their manuscript.
Author Response

(The authors gave the same response as above.)

Round 2
Reviewer 1 Report
The authors answered most of the questions as stated and did a good job for the most part. The manuscript appears improved. However, there are still some points that need to be changed if the authors didn't want to make changes before:
1. I'll quote the entire section to make it clearer:
"4.3 Experimental Design.
Experiment 1 was designed to develop and characterise an in vitro neurosphere model to study adult SVZ neurogenesis. This model allowed us to test the efficacy of D-Pen in attenuating Cu-induced neurosphere disruption and the underlying mechanisms.
In experiment 2, our in vitro studies were extended to in vivo to establish an ICV infusion animal model to investigate how local administration of D-Pen by ICV infusion alters Cu concentration in the brain and to observe subsequent adult neurogenesis along the SVZ-RMS-OB axis in response to short- and long-term D-Pen infusion at two doses. In the long-term infusion, we also examined the changes in homeostasis of the NSPC pool and the expression patterns of CTR1 and MT3 by IHC, two important Cu-regulatory proteins, in the SVZ.
Finally, Experiment 3 examined whether D-pen infusion alters the expression of CTR1 and MT3 in the choroid plexus (CP), a blood-rich tissue in the brain ventricles near the SVZ that regulates Cu transport in the CSF [50]."
From this section, it looks like Experiment 3 is a completely different experiment, i.e., it doesn't look like it was also done in vivo and applied with ICV D-Pen infusion!The explanation should be added so there is no mistunderstanding while reading it. So if you want to leave it as experiment 3, at least explain it in one sentence...something like this...
"Experiment 3 also extended our in vitro studies to in vivo to investigate whether D-pen infusion alters the expressions of CTR1 and MT3 in the choroid plexus (CP), a blood-rich tissue in the brain ventricles near the SVZ that regulates Cu transport in the CSF [50]."
2. Why the authors did not include an image with Brdu/DCX before Figure1H?
It was never the question asked for Fig1E-G, only for Fig1H. They answered in first revision: "However, in Fig1H, neurospheres were induced to differentiate into neurons by removing growth factors; therefore, NeuN, a marker for mature neurons, was detected."
And why not DCX/Brdu?
3.The authors have good intention to make this figure clearer and more visible, but unfortunately the arrows are too small (the arrows in Figure 8 are better). It seems like there are some spots or artefacts on figures. Please change arrows (make them larger) so that the arrows are clearly visible, because when I enlarge the figure the arrows loose good resolution.
4. The authors didn't made any corrections of magnification-pictures in the photoshop as I strictly asked to do so and not to leave them in this native form with lines for figure 8. Again, the overlapping signal within this cell in fig8C (the magnification figure) cannot be seen as yellow. This should be visible, and is not.
5. The differences between the D-pen treatment and the control in fig12 A (CTR -1) should be better presented. This is still not as good. These representative pictures are not in good relationship with results presented in belonging graph (Fig12c). At least marked this with visible arrows.
